Nutritional content of Totoaba macdonaldi (Gilbert, 1890), Antioxidants and lipid peroxidation in muscle

Conde-Guerrero Priscila 1 2
Méndez-Rodríguez Lia C. 2
de Anda-Montañez Juan A. 3
Zenteno-Savín Tania tzenteno04@cibnor.mx 2
1 Biología Marina, Universidad Autónoma de Baja California Sur , La Paz , Baja California Sur , México
2 Programa de Planeación Ambiental y Conservación, Centro de Investigaciones Biológicas del Noroeste, S.C. , La Paz , Baja California Sur , México
3 Programa de Ecología Pesquera, Centro de Investigaciones Biológicas del Noroeste, S.C. , La Paz , Baja California Sur , México
Okpala Charles
Electronic publication date: 2021 Mar 31
Publication date: 2021
Volume: 9
Electronic Location ID: e11129
Received 2020 Oct 27; Accepted 2021 Feb 27
Copyright: ©2021 Conde-Guerrero et al.
Copyright year: 2021
Copyright holder: Conde-Guerrero et al.
License: This is an open access article distributed under the terms of the Creative Commons Attribution License, which permits unrestricted use, distribution, reproduction and adaptation in any medium and for any purpose provided that it is properly attributed. For attribution, the original author(s), title, publication source (PeerJ) and either DOI or URL of the article must be cited.
License URL: https://creativecommons.org/licenses/by/4.0/

Keywords: Totoaba, Antioxidants, Peroxidation, Nutrition, Fish

Funding: SEP-CONACYT 2011-01/165376 CIBNOR (EP, PPAC) This study was funded by SEP-CONACYT (2011-01/165376) and CIBNOR (EP, PPAC). The funders had no role in study design, data collection and analysis, decision to publish, or preparation of the manuscript.

==============================
Background

Totoaba, Totoaba macdonaldi, is an endemic species of the Gulf of California, where wide variations in sea temperature throughout the year, surface salinities that gradually increase towards the north, and contamination by discharge of wastewater have been recorded. In addition to the challenges of reproduction and swimming, its characteristic biannual migration presents totoaba with changes in environmental factors that could affect oxidative stress indicators. The objective of this study was to assess spatial and seasonal changes in the oxidative stress indicators in muscle samples of totoaba.

Methods

Reactive oxygen species production, antioxidant enzyme activities and lipid peroxidation levels were quantified by spectrophotometry.

Results

Results suggest spatial-temporal variations of the oxidative stress indicators in muscle of totoaba that may be associated to a complex interaction between environmental and biological factors, including reproduction and nutrient availability. These results contribute to explain the appeal of totoaba as a marketable meat and suggest totoaba may provide antioxidant nutrients to consumers.

Introduction

Totoaba, Totoaba macdonaldi (Gilbert, 1890), is an endemic species of the Gulf of California, characterized by its longevity and size, reaching up to two meters in length and over 100 kg body mass (Cisneros-Mata, Botsford & Quinn, 1997; Guevara, 1990; Román Rodríguez & Hammann, 1997). Totoaba tend to aggregate for reproduction during the spawning season in the Upper Gulf of California (Flanagan & Hendrickson, 1976; De Anda-Montañez et al., 2013), which renders them vulnerable to fishing. Although commercial fishing of totoaba has been banned for over 40 years by Mexican authorities (DOF, 2010) and international trade is forbidden by international organizations (CITES, 2010; IUCN, 2016), the high demand for dried swimming bladders (known as maw) in Asia, particularly in China, with prices that can achieve US$ 5,000 or more per kilogram, ex-vessel price, has increased the illegal fishing of totoaba with gillnets, mainly in the Upper Gulf of California (Morell, 2017; Valenzuela-Quiñonez et al., 2015). Simultaneously, the bycatch of the critically endangered vaquita (Phocoena sinus) associated with totoaba illegal fishing is an international concern (Morell, 2017) that must be properly addressed in a holistic manner based on scientific information, but that is beyond the scope of this study.

Totoaba is a traditional staple food for people living in coastal towns in the upper Gulf of California. This species has been recently subjected to aquaculture (mariculture) (True, Silva Loera & Castro Castro, 1997; Juarez, Konietzko & Schwartz, 2016) and is available as a main course in several restaurants in Mexico and the USA (Wright, 2017). Studying the antioxidant defense in wild fish provides a basis of the oxidative state of the fresh product which is later subjected to post-mortem handling processes (storage, distribution, or heating during cooking) that promote additional oxidative damage. Totoaba muscle contains an elevated amount of Ω-6 HUFAs; hence, it is expected to be susceptible to lipid and protein oxidation, which may affect its organoleptic properties.

Assessing oxidative stress in fish muscle provides valuable information. Reactive oxygen species (ROS) production, antioxidant content and, thus, oxidative damage, in animals, including fish, varies in response to both environmental and physiological factors (Halliwell & Gutteridge, 2007; Lushchak, 2011b). Sea temperature, dissolved oxygen, salinity, presence of contaminants, photoperiod, among others, are environmental factors that can influence the antioxidant status of fish (Lushchak, 2011b). Nutritional status, physical activity, age, sex, as well as reproduction itself, could also contribute to variations in tissue antioxidant content (Halliwell & Gutteridge, 2007). Diet, physical activity and reproduction can alter fish metabolism and oxidative stress indicators (Filho, Giulivi & Boveris, 1993; Aleshko & Lukyanova, 2008; Birnie-Gauvin et al., 2017). Antioxidants neutralize ROS, contributing to avoid oxidative damage, and thus oxidative stress, in cells and tissues; oxidative stress has been associated to cardiovascular and respiratory diseases, cancer, immune deficiency, some inflammatory conditions, among others (Halliwell & Gutteridge, 2007). Evidence suggests that including antioxidants in the diet contributes to maintain human health (Kulawik et al., 2013).

Totoaba perform two annual migrations; during summer months the organisms move away from the coast towards deeper and colder waters, and in winter-spring adults swim towards the Colorado River delta to reproduce (Cisneros-Mata, Botsford & Quinn, 1997; De Anda-Montañez et al., 2013). The Gulf of California is characterized by wide variations in sea temperature throughout the year, high surface salinities that gradually increase towards the north, and contamination by discharge of wastewater, particularly in the upper Gulf of California. Therefore, totoaba is faced with seasonal and spatial changes in both environmental and biological factors throughout its lifespan which affect the quality of their muscle. Muscle is the main fish tissue consumed by humans. The aim of this study was to analyze and compare the antioxidant enzyme activities and lipid peroxidation levels in muscle of totoaba collected in different sites and seasons along its migratory route, towards assessing the potential variation in nutritional content depending on the season and location the fish was caught.

Materials & Methods

Overview of experimental study

Totoaba macdonaldi organisms were collected according to a stratified random sampling during 11 field trips in different locations and dates around the Gulf of California (Fig. 1). From each specimen, muscle samples were dissected, frozen, and transported to the laboratory, where they were kept frozen until analyzed.

Figure 1 Totoaba (Totoaba macdonaldi) study area in the Gulf of California.

Sample collection and preparation

Totoaba were collected from the Upper Gulf of California in the three types of habitats (estuarine, Upper Gulf of California and Colorado River Delta Biosphere Reserve; rocky, area around Consag and Las Encantadas Islands; continental, along the coasts of Sonora and Sinaloa) where the fish inhabit (Fig. 1). Field trips were conducted in April, May, and November of 2010; February, March, October, November, and December of 2011; April 2012; and January, February of 2013 to collect samples during all seasons. Totoaba were found in the estuarine habitat (spawning grounds; De Anda-Montañez et al., 2013) in winter and spring, in the coastal and rocky habitats (feeding areas; De Anda-Montañez et al., 2013) in autumn, but were not found in any of the sampled sites in summer. In summer, when sea surface temperatures in the Gulf of California can be over 28 °C, totoaba migrate to deeper (∼70 m) colder (∼21 °C) zones (Hernández-Tlapale et al., 2020) being, thus, inaccessible to fishing gear. Fishes were caught with an empirical gill net (built following the knowledge and experience of the fishermen) of approximately 120 m long, 4 m wide (depth) and 10″mesh opening, as well as with individual fishing rods with number 5 hook. Once obtained, weight and length were recorded, and the fish was dissected. Sex was identified by histological criteria (De Anda-Montañez et al., 2013). Organisms that presented gonadal development from stage III of their reproductive cycle were classified as mature (De Anda-Montañez et al., 2013). Those cases in which gonadic development did not allow for sex determination were grouped as undifferentiated. Data were grouped according to habitat (estuarine, rocky, continental), collection date (spring, autumn, winter), and sex and stage of reproductive maturity (immature females (FI), mature females (FM), immature males (MI), mature males (MM) and undifferentiated (U)). Sampling was performed under scientific collection permits issued by the Mexican government (SGPA/DGVS/02913/10, SGPA/DGVS/05508/11 and SGPA/DGVS/00039/13), as described in the research protocol of the project entitled “Health and Conservation status of the totoaba population (Totoaba macdonaldi) in the Gulf of California: a critically endangered species” (CONABIO: FB1508/HK050/10; CONACYT 2011-01/165376), and according to the institutional ethical guidelines for critically endangered species (CIBNOR, SGPA/DGVS, CONABIO, CONACYT).

All muscle samples (n = 174) collected were immediately frozen by immersion in a 10-liter cryoshipper (Thermo Fisher Scientific, Waltham, MA, USA) with liquid nitrogen and transported to the Oxidative Stress Laboratory at CIBNOR, La Paz, Baja California Sur, Mexico, where they were kept frozen (−80 °C) until the corresponding analyses were performed. From each muscle sample, 100 mg of tissue was taken and homogenized (50 mM phosphate buffer, pH 7.5, 1 mM phenylmethylsulfonyl fluoride (PMSF)). The homogenized samples were centrifuged for 15 min at 2124× g and the supernatants obtained were immediately analyzed. The reagents used were obtained from Sigma-Aldrich Chemical Co. (San Luis, MO) and Bio-Rad Laboratories (Hercules, CA).

Superoxide radical production

Superoxide radical (O2⋅−) production was analyzed as an indicator of ROS production according to Drossos et al. (1995), based on its production rate during the reduction of ferricytochrome c. Muscle samples were placed in Krebs buffer (0.11 M NaCl, 0.0047 M KCl, 0.012 M MgSO4, 0.012 M NaH2PO4, 0.025 M NaHCO4 and 0.001 M glucose). Cytochrome c (15 µM) was added to each sample, which was incubated at 37 °C in a rocking shaker for 15 min. N-ethylmaleimide (3 mM) was added to stop the reduction of cytochrome c. Samples were centrifuged at 2124× g at 4 °C for 10 min and the supernatant was transferred to a cell to measure the absorbance in a spectrophotometer (Beckman Coulter DU 800 UV / Visible, Fullerton, CA) at 550 nm wavelength. All samples were analyzed in triplicate. The production of O2⋅− was calculated based on the extinction coefficient E550 = 21 nM L−1 cm−1. Superoxide radical production data were expressed in nmol per min per mg of protein.

Lipid peroxidation

Lipid peroxidation levels were evaluated as an indicator of oxidative damage and meat quality following the method described by Persky et al. (2000). Lipid peroxidation was analyzed as the content of thiobarbituric acid reactive substances (TBARS). A standard curve was prepared with a solution of 1,1,2,3-tetraethoxypropane (TEP) in a range of 0 to 5 nmoles 250 µL−1. The samples and the standard curve were incubated at 37 °C with constant agitation for 15 min. Upon completion, trichloroacetic acid (TCA, 0.76 M in 1 M HCl) was added to stop the reaction. Thiobarbituric acid (TBA, 1%) was added and the tubes were incubated at 90 °C for 10 min in a constant shaking water bath. Subsequently, they were placed in an ice bath and centrifuged (2124× g) for 10 min at 4 °C. The supernatant was recovered, and absorbance was read in a spectrophotometer at 530 nm wavelength. All samples were analyzed in triplicate. The results were calculated from the standard curve and expressed in nanomoles of TBARS per mg of protein.

Antioxidant enzymatic activity

Superoxide dismutase (SOD, E.C.1.15.1.1) activity was measured by calculating the inhibition of nitroblue tetrazolium (NBT) reduction and expressed in units of SOD per mg of protein according to the method of Suzuki (2000). One unit of SOD activity is defined as the amount of enzyme needed to inhibit 50% of the maximum reaction of O2⋅− with NBT. SOD activity is expressed in U mg−1 protein. Catalase (CAT, E.C.1.11.1.6) activity was quantified by following the decrease of hydrogen peroxide (H2O2) content at 240 nm wavelength (Aebi, 1984) and was expressed in units of CAT per mg of protein. One unit of CAT activity is defined as the amount of enzyme that catalyzes the decomposition of 1 µmol of H2O2 per minute. CAT activity is expressed in U mg−1 protein. Glutathione peroxidase (GPx, E.C.1.11.1.9) activity was measured according to Flohé & Günzler (1984), this method uses H2O2 as a substrate coupled with the oxidation of nicotinamide adenine dinucleotide phosphate (NADPH) catalyzed by glutathione reductase (GR) at 240 nm wavelength. One unit of GPx is defined as the amount of enzyme that oxidizes 1 µmol of NADPH per minute. The activity of GR (E.C. 1.6.4.2) was measured by following the decreasing levels of NADPH according to the method of Goldberg & Spooner (1983). Results were expressed in units of GR per mg of protein. One unit of GR is defined as the amount of enzyme that oxidizes 1 µmol of NADPH per minute. The enzymatic activity of glutathione S-transferase (GST, E.C.2.5.1.18) was determined following the formation of the thioether product in the reaction between glutathione and 1-chloro, 2,4-dinitrobenzene (CDNB) according to the method of Habig & Jakoby (1981). Enzymatic activity was expressed in units of GST per mg of protein. One unit of GST activity is defined as the amount of enzyme that catalyzes the conjugation of 1 µmol of CDNB per minute at 25 °C. All samples were analyzed in triplicate.

Soluble proteins

To standardize the enzymatic activities, the concentration of proteins present in the tissue extract was quantified following the method of Bradford (1976) using commercial kits (BioRad®). Samples were diluted 1:50 with phosphate buffer (50 mM, pH 7.5, EDTA). A standard curve of bovine serum albumin (BSA) was prepared in a concentration range of 0–2 mg mL−1. In a microplate, distilled water (dH2O), Bradford dye and the BSA standard curve or sample were added to each well. In addition, a blank was prepared with dH2O and Bradford dye only. After 15 min, the absorbance at 620 nm wavelength was read on the microplate reader (Multiskan FC, Thermo Scientific, Finland). All samples were analyzed in triplicate. The protein content in each sample was calculated based on the regression of the standard curve. Data are expressed in mg of protein per mL.

Statistical analysis

Tests were performed to determine if the dataset follows the assumptions of normality and homoscedasticity. Non-parametric Kruskal-Wallis tests were applied to determine significant differences in O2⋅− production rate, TBARS levels and antioxidant enzyme activities between seasons, habitats, and sex/maturity stage. The significance level (α) of 5% (p = 0.05) was taken to denote statistical differences for all tests. The statistical analyses were performed with STATISTICA® (StatSoft, Inc., 2002).

Two generalized linear models (GLMs) were created to identify the variables with the greatest contribution to the variability of oxidative damage, as a proxy for meat quality, in totoaba muscle samples. Each model was developed according to the activity of each specific enzyme. Superoxide radical (O2⋅−) production rate and lipid peroxidation (TBARS) levels were selected as the continuous dependent variables (response); while season, habitat and sex/maturity stage were considered the categorical explanatory variables, and the activity of each enzyme was included as a continuous explanatory variable (Table 1). The models assumed the Gamma distribution because it yielded the best fit, as it was not significantly different from the Chi-square test (p > 0.05). The link function used in all models was Log. The models were built using a forward procedure, that is, explanatory variables were added to the null model, and the best model was selected based on Akaike’s information criterion (AIC), bayesian information criterion (BIC) and residual deviance (RD). This selection process was repeated for both models, the superoxide radical (O2⋅−) production rate and lipid peroxidation (TBARS) levels. The models were validated by visual analysis of the residuals and observed/predicted values. The best model for each biomarker was selected based on AIC.

Table 1 Description of possible variables affecting the oxidative stress indicators in the muscle of Totoaba macdonaldi, captured in the Gulf of California, Mexico.

Variable	Type	Description	
Season	Categorical	Spring: Spring 2010, 2011, 2012
Autumn: Autumn 2010 and 2011
Winter: Winter 2011, 2013	
Habitat	Categorical	Habitat 1: Estuaries zone,
South of San Felipe
Habitat 2: Rocky area,
Encantadas islands, Consag island
Habitat 3: Continental area,
Lobos Bay, El desemboque (Seris), Las Grullas (Sinaloa)	
Sex/reproductive maturity status	Categorical	FI: immature female
FM: mature female
MI: immature male
MM: mature male
U: undifferentiated	
Superoxide dismutase (SOD)	Continuous	Enzymatic activity (U mg−1 protein)	
Catalase (CAT)	Continuous	Enzymatic activity (U mg−1 protein)	
Glutathione peroxidase (GPx)	Continuous	Enzymatic activity (U mg−1 protein)	
Glutathione reductase (GR)	Continuous	Enzymatic activity (U mg−1 protein)	
Glutathione S-transferase (GST)	Continuous	Enzymatic activity (U mg−1 protein)	

Results

A total of 174 totoaba individuals were analyzed. Of these, 64 were FI, 25 FM, 54 MI, 21 MM and 10 U. Comparisons of the antioxidant enzyme activities and oxidative damage levels were made between the different habitats, seasons, and sex/maturity stage (Table 2). Not all sex/maturity stage categories were found in all seasons, nor habitats, perhaps due to the combination of the species migratory (horizontal and vertical) behavior and the specific environmental conditions at each site and season (Fig. 2). In the estuaries, higher O2⋅− production rate and lower activities of CAT, GPx and GST were observed for the FM in winter compared to spring (p < 0.05). No significant differences were observed in TBARS levels (p > 0.05). In the rocky habitat, higher TBARS levels and GPx activity, as well as lower SOD activity, were observed for the FI in the spring compared to autumn (p < 0.05). In the continental habitat, higher GST activity was observed for the MI in winter than in autumn (p < 0.05). In spring, the FM had higher GST activity, while the MM had lower O2⋅− production rate and higher GPx activity in the estuaries than in the rocky areas (p < 0.05). In autumn, both the MI and U totoaba had higher GST activity in the rocky areas as compared to the continental areas (p < 0.05). In winter, the FI had lower O2⋅− production rate and higher GPx activity in the estuaries than in the rocky areas (p < 0.05), while MI had lower antioxidant enzyme activities in the estuaries than in the continental areas (p < 0.05). The FI had higher GPx activity than the FM totoaba in spring and lower GST activity than the U totoaba in autumn in the rocky areas (p < 0.05). The MI had higher GST activity than the U totoaba in autumn and higher TBARS levels, and GR and GST activities than FI totoaba in winter in the continental areas (p < 0.05).

Table 2 Oxidative stress indicators in muscle samples of totoaba (Totoaba macdonaldi) in the Gulf of California.

		Variable	FI	FM	MI	MM	U	
Estuaries	Spring	O2⋅−	–	0.00009 (0.00007–0.0002)*	–	0.00010 (0.00008–0.00011)+	–	
TBARS	–	1.22 (0.86–2.26)	–	0.61 (0.37–1.15)	–	
SOD	–	8.43 (6.10–16.37)	–	8.15 (6.33–24.43)	–	
CAT	–	17.27 (7.33–19.04)*	–	8.65 (4.50–21.82)	–	
GPx	–	0.57 (0.30–0.79)*	–	0.07 (0.05–0.31)+	–	
GR	–	0.20 (0.07–0.36)	–	0.11 (0.09–0.32)	–	
GST	–	0.03 (0.02–0.05)*+	–	0.02 (0.01–0.06)	–	
Winter	O2⋅−	0.00012 (0.00008–0.00018)	0.0004 (0.0003–0.0006)*	0.0002 (0.0001–0.0004)	0.0005 (0.0002–0.0006)	–	
TBARS	0.23 (0.15–0.83)	0.73 (0.44–1.15)	0.39 (0.30–0.93)	0.73 (0.49–1.31)	–	
SOD	4.90 (3.33–11.23)	12.91 (4.58–18.80)	4.91 (2.66–9.26)+	12.74 (6.99–31.56)	–	
CAT	3.40 (1.89–5.40)	4.45 (2.42–5.59)*	6.91 (0.84–7.76)+	3.79 (2.92–5.04)	–	
GPx	0.10 (0.10–0.15)	0.12 (0.08–0.21)*	0.19 (0.11–0.24)+	0.17 (0.09–0.24)	–	
GR	0.07 (0.05–0.16)+	0.20 (0.11–0.24)	0.08 (0.06–0.12)+	0.18 (0.12–0.36)	–	
GST	0.003 (0.002–0.024)	0.006 (0.004–0.015)*	0.004 (0.002–0.015)+	0.011 (0.005–0.040)	–	
Rocky area	Spring	O2⋅−	0.0003 (0.0002–0.0008)	0.0001 (0.0001–0.0008)	0.0005 (0.00005–0.0008)	0.0003 (0.0002–0.0013)+	–	
TBARS	0.88 (0.59–1.09)*	1.24 (0.78–2.41)	1.23 (0.70–2.03)	0.53 (0.18–1.46)	–	
SOD	6.72 (4–8.09)*	11.25 (9.27–14.34)	5.50 (4.80–14.85)	22.43 (10.93–26.33)	–	
CAT	6.40 (3.09–11.86)	5.00 (3.83–10.58)	11.26 (7.76–36.61)	14.94 (6.17–21.45)	–	
GPx	1.09 (0.57–1.61)*b	0.18 (0.11–0.48) a	0.56 (0.37–1.34) ba	1.58 (0.57–2.35)+b	–	
GR	0.37 (0.26–0.65)	0.18 (0.17–0.21)	0.22 (0.17–0.76)	0.40 (0.20–0.64)	–	
GST	0.03 (0.01–0.06)	0.02 (0.01–0.02)+	0.01 (0.01–0.04)	0.04 (0.02–0.06)	–	
Autumn	O2⋅−	0.0004 (0.0002–0.0005)	–	0.0004 (0.0002–0.0008)	–	0.0004 (0.0002–0.0009)	
TBARS	0.44 (0.24–0.74)*	–	0.49 (0.28–0.85)	–	0.52 (0.46–1.12)	
SOD	11.71 (6.62–19.83)*	–	14.07 (4.69–21.06)	–	16.52 (7.40–33)	
CAT	8.49 (5.25–16.67)	–	8.00 (5.52–15.93)	–	40.88 (21.83–53.16)	
GPx	0.31 (0.13–0.58)*	–	0.30 (0.09–0.51)	–	0.40 (0.11–1.60)	
GR	0.19 (0.12–0.39)	–	0.27 (0.12–0.50)	–	0.24 (0.16–0.67)	
GST	0.02 (0.01–0.04)b	–	0.02 (0.01–0.05)+ b	–	0.07 (0.05–0.09)+a	
Continental area	Autumn	O2⋅−	–	–	0.0002 (0.0002–0.0003)	–	0.0002 (0.0001–0.0004)	
TBARS	–	–	0.22 (0.19–1.99)	–	0.36 (0.20–0.63)	
SOD	–	–	4.90 (3.51–16.82)	–	10.66 (3.15–19.35)	
CAT	–	–	0.006 (0.004–0.011)	–	2.90 (2.58–6.28)	
GPx	–	–	0.26 (0.14–0.48)	–	0.23 (0.13–0.51)	
GR	–	–	0.16 (0.09–0.33)	–	0.18 (0.10–0.22)	
GST	–	–	0.006
(0.004–0.011)*+a	–	0.0010
(0.0006–0.0019)+b	
Winter	O2⋅−	0.0002 (0.0001–0.0004)	–	0.0004 (0.0001–0.0005)	–	–	
TBARS	0.34 (0.21–0.77)a	–	1.28 (0.71–1.52)b	–	–	
SOD	13.50 (7.43–21.72)	–	23.71 (17.95–30.46)+	–	–	
CAT	10.66 (5.03–16.39)	–	15.35 (9.36–48.43)+	–	–	
GPx	0.36 (0.13–0.66)	–	0.68 (0.50–1.04)+	–	–	
GR	0.26 (0.23–0.32)+a	–	0.35 (0.33–0.57)+b	–	–	
GST	0.013 (0.008–0.030)a	–	0.04 (0.03–0.08)*+b	–	–	
Notes.

O2- Superoxide radical production rate (nmol min−1 mg−1 protein)

TBARS concentration of thiobarbituric acid reactive substances (nmol mg-1 protein)

SOD superoxide dismutase

CAT catalase

GPX glutathione peroxidase

GR glutathione reductase

GST glutathione Stransferase, antioxidant enzyme activities (Units mg-1 protein)

Data are expressed as median and percentiles (25–75%). FI, Immature females (n = 64), FM, Mature females (n = 25), MI, Immature males (n = 54), MM, Mature males (n = 21), U, Undifferentiated (n = 10).

* Significant differences between seasons.

+ Significant differences between habitats.

Different letters denote significant differences between sex/reproductive maturity status groups for each season and habitat. Level of statistical significance, p < 0.05.

The results of the selection process and the parameters obtained for the GLMs for O2⋅− production rate and TBARS levels are shown in Tables 3 and 4, respectively. The best-fit model selected based on AIC and BIC to explain the O2⋅− production rate included enzymatic (GR and SOD) activity, the habitat (estuarine, rocky and continental areas), as well as the season (spring, autumn and winter). This model started with 17 parameters (k = 17), but only 7 of these (k = 7), intercept, GR and SOD activities, estuarine and rocky areas, and spring, as well as the interaction between habitat and season, were significant (p < 0.05) in explaining the O2⋅− production rate. The best model selected based on AIC and BIC for lipid peroxidation (TBARS) levels included enzymatic (GST, CAT, SOD and GPX) activity and season (spring, autumn and winter). This model started with 17 parameters (k = 17), but only four (k = 4), intercept, SOD, spring and autumn, significantly contributed to explain the variation in TBARS levels. In addition, the best-fit model for each indicator was validated using analysis of residuals (Fig. 3), which suggests that the variance of the residuals is homogeneous over the independent variables in the models; thus, confirming that the models fit the data reasonably well.

Figure 2 Number of samples of totoaba (Totoaba macdonaldi) collected in the Gulf of California during 2010–2013.

FI, Immature females, FM, mature females, MI, immature males, MM, Mature males, U, Undifferentiated.

Table 3 Generalized linear models to describe the superoxide radical (O2-) production rate and lipid peroxidation (TBARS) levels (results not shown) in muscle of the totoaba, Totoaba macdonaldi.

Models were constructed following the process shown. Best-fit models, shown in bold, were selected based on Akaikes, information criterion (AIC), Bayesian information criterion (BIC) and residual deviance (RD).

Modelsuperoxide radical (O2⋅−) production	AIC	BIC	RD	
Null
SOD
CAT
GPX
GR
GST
Habitat
Season
Sex/Maturity
GR + SOD
GR + CAT
GR + GPX
GR + GST
GR + Habitat
GR + Season
GR + Sex/Maturity
GR + SOD + CAT
GR + SOD + GPX
GR + SOD + GST
GR + SOD + Habitat
GR + SOD + Season
GR + SOD + Sex/Maturity
GR + SOD + Habitat + CAT
GR + SOD + Habitat + GPX
GR + SOD + Habitat + GST
GR + SOD + Habitat + Season
GR + SOD + Habitat + Sex/Maturity
GR + SOD + Habitat + Season + CAT
GR + SOD + Habitat + Season + GPX
GR + SOD + Habitat + Season + GST
GR + SOD + Habitat + Season + Sex/Maturity	−2327
−2309
−2247
−2288
−2315
−2249
−2332
−2325
−2321
−2291
−2233
−2283
−2260
−2319
−2315
−2308
−2207
−2259
−2233
−2297
−2290
−2283
−2213
−2265
−2239
−2299
−2282
−2216
−2265
−2243
−2284	−2321
−2299
−2238
−2279
−2306
−2239
−2320
−2313
−2302
−2278
−2221
−2271
−2247
−2303
−2299
−2286
−2192
−2243
−2218
−2278
−2272
−2259
−2192
−2243
−2218
−2271
−2235
−2186
−2234
−2212
−2221	152
141
143
145
138
144
145
150
151
132
131
136
134
133
136
138
126
130
130
125
130
132
119
123
123
120
124
113
119
116
116	
Notes.

SOD superoxide dismutase

CAT catalase

GPx glutathione peroxidase

GR glutathione reductase

GST glutathione S-transferase, antioxidant enzyme activities (Units mg−1 protein)

Selected best models are shown in bold.

Table 4 Statistics of goodness of fit, parameter estimates and standard error (SE) from model variables in the final fitted general linear models (GLM).

Models	O2⋅−		TBARS	
Error
Link
N
RD
DF	Gamma
Log
172
120
158	Gamma
Log
170
80
147	
Level of effect	Estimate	SE	Level of effect	Estimate	SE	
Intercept
GR
SOD
Estuaries
Rocky
Spring
Autumn
Habitat*Season	−8.584
1.042*
0.017*
−0.293*
0.719*
−0.411*
−0.028
−0.374*	0.134
0.328
0.007
0.175
0.147
0.138
0.124
0.173	Intercept
SOD
CAT
GST
GPX
Spring
Autumn
–	−0.674*
0.015*
0.011
4.617
0.020
0.447*
−0.385*
–	0.107
0.006
0.006
3.15
0.139
0.099
0.080
–	
AIC
BIC	−2299
−2271		182
206	
Notes.

n number of data

RD residual deviance

DF degrees of freedom

AIC Akaikes information criterion

* Values were significantly different (p < 0.05) from the intercept.

Figure 3 Residual analysis for superoxide radical (O2-) production rate (A–B) and lipid peroxidation (TBARS) levels (C–D).

The best fit models were selected according to Aikaike’s information criterion (AIC) and bayesian information criterion.

Discussion

The spatial–temporal variation of the oxidative stress indicators in fish suggests a complex interaction between different environmental and biological factors (Wilhelm-Filho et al., 2001; Kopecka & Pempkowiak, 2008; Aras et al., 2009; Radovanović et al., 2010; Pavlović et al., 2010). Results from this study suggest that the relative antioxidant content and oxidative damage may vary depending on the geographic location and the season when the products are obtained due to the variation in conditions wild totoaba, T. macdonaldi, face along its distribution. Furthermore, these indicators provide relevant information about fish product quality, and may be an important reference for human health (Tripathy, 2016).

Fish products are at risk of quality loss due to oxidation; lipid peroxidation can lead to a rancid taste and development of many compounds that may cause adverse effects to human health (Tripathy, 2016). Some peroxidation products, including aldehydes, can react with specific amino acids and form carbonyls and other protein aggregates. These secondary products may cause additional nutritional loss (Ahmed et al., 2016). Higher O2⋅−production rate and lower activities of CAT, GPx and GST were observed in winter compared to spring in muscle samples of the FM totoaba collected in estuaries. Reduced activity of antioxidant enzymes has been reported at low temperatures (Viarengo et al., 1991; Speers-Roesch & Ballantyne, 2005; Lushchak, 2011a). In the Upper Gulf of California, the sea surface temperature has a wide range of variation and can decrease to 10 °C in winter (Soto-Mardones, Marinone & Parés-Sierra, 1999). Low temperature in winter could be associated to the lower antioxidant enzyme activities in the muscle of FM totoaba in this study.

Active swimming, associated with migration in some species, can also contribute to increased ROS production due to increased oxygen consumption by red muscle mitochondria (Costantini, 2008; Amérand et al., 2010; Mortelette et al., 2010; Wilson et al., 2014). Totoaba is a migratory species with markedly seasonal reproduction; that is, the development of the gonads begins during the winter and continues during spring (De Anda-Montañez et al., 2013). During the months of March and April, the largest number of brood stock is reported in the estuaries; therefore, an increase in O2⋅− production rate could be expected in totoabas in estuarine areas, assuming that they have a larger metabolic activity. However, lower O2⋅− production rate and GPx activity were observed in muscle samples of the MM totoabas in the estuaries during spring, as compared to the rocky habitat. It is possible that, during the reproductive season, most of the resources (energy, nutrients, antioxidant defenses, etc.) are diverted preferentially to the gonads; under these circumstances (lower availability of resources for maintenance), other tissues, such as muscle, may become susceptible to oxidative stress and potential loss of meat quality (Wiersma et al., 2004; Smith et al., 2014).

The FI totoabas at the rocky habitat had higher TBARS levels and GPx activity, but lower SOD activity, in spring as compared to autumn. Because these are immature fish, the observed differences may be associated to diet and nutrition. During winter and spring, higher primary productivity has been reported in the northern part of the Gulf of California (Pérez-Arvizu, Aragón-Noriega & Espinosa-Carreón, 2013); which suggests greater food abundance and diversity for the organisms that live there, including totoaba. Although juvenile totoaba consume diets with high protein levels and moderate lipid content, their natural diet seems to be rich in Ω-3 and Ω-6 highly unsaturated fatty acids (HUFAs) (López et al., 2006). An increase in lipid content increases the potential for generation of lipid radicals and, thus, the susceptibility to oxidation, as has been reported for fish and sea urchins (Malanga et al., 2007; Zhang et al., 2007). Although no differences were observed in the O2⋅− production rate in FI totoabas in this study, other ROS not quantified in this study, such as H2O2, could contribute to lipid peroxidation; this would coincide with the antioxidant enzyme activities observed in FI totoabas.

Fish products are an important part of a healthy diet, as they are sources of several nutrients, including antioxidants. Increasing evidence of the benefits of dietary antioxidants in human health include lowering the risk of some cardiovascular diseases, aging and even cancer (Kulawik et al., 2013). The levels of endogenous antioxidant enzymes contribute to maintain the fish meat quality, to prolong shelf-life of products and to decrease the health risks from oxidation to the fish, the fish products, and consumers (Sen & Mandal, 2016).

The TBARS levels, as well as GPx and GST activities were higher in the MI compared to the FI in winter in the continental habitat. This relative difference could be attributed to antioxidant defenses not quantified in this study, such as vitamin A, vitamin C, vitamin E, carotenoids, glutathione, flavonoids, or estrogens (Viña et al., 2005; Amérand et al., 2010). However, these differences between sexes should not be generalized, since no differences were observed between FI and MI in other seasons or habitats, (e.g., estuaries), nor between sexes for mature organisms.

Higher activity of SOD, CAT, GPx, GR and GST was observed in fish from the continental habitat as compared to those in the estuaries during winter in MI totoaba. This may be a response to environmental changes due to the upwelling phenomenon that occurs on the eastern coast of the Gulf of California in winter, which implies a cooler, nutrient-rich water transport to the surface (Lavín & Marinone, 2003; Pérez-Arvizu, Aragón-Noriega & Espinosa-Carreón, 2013). The greater availability of nutrients in the water column has been associated to greater food availability and fish nutritional status. It is possible that the increased activity of all the antioxidant enzymes quantified in MI totoaba in this study is a response to the increased metabolic rate that could be implied by the greater food availability. Alternatively, the variations in the activity of the antioxidant enzymes observed in the present study could be due to other environmental factors, such as pollution. Greater GST activity was observed in the muscle samples of the (MI, U) totoabas in the rocky habitat compared to those in the continental habitat in the autumn. GST participates in the detoxification of a number of xenobiotic compounds and its increased activity has been reported in many fish species (Porte et al., 2000; Lopes et al., 2001; Napierska & Podolska, 2005; Kopecka & Pempkowiak, 2008). It has been suggested that the main entry of contaminants (organochlorine pesticides, trace elements) into the Upper Gulf of California is the discharge of wastewater from the agricultural valleys of Mexicali and San Luis Río Colorado in the delta area. Shumilin et al. (2002) reported an intense resuspension of sediments in the delta with a subsequent transport to the south; the concentrations of most of the elements studied are higher in the southwest margin of the Upper Gulf. Possibly, the tidal processes that cause the transport of sediments towards the south could also participate in the transport of other pollutants.

The results of the GLM analyses suggest that the variables that best explained the variability of the oxidative damage in muscle of totoaba are the antioxidant enzyme activities, habitat (specifically, the rocky area of the Consag and Las Encantadas Islands) and season (spring). It is possible that the variation in O2⋅− production rate and TBARS levels were related to the environmental changes that occur along the Gulf of California, spatially and temporarily. Extreme environmental conditions occur in the northern Gulf of California, especially the delta area. Variations in the oxidative stress indicators in response to changes in biotic and abiotic factors have been previously reported in several fish species (Aras et al., 2009; Pavlović et al., 2010; Oliva et al., 2012), including T. macdonaldi (Hernández-Aguilar et al., 2017). Results from this study suggest that totoaba faces spatial–temporal variation in muscle oxidative stress indicators that may be associated to a complex interaction between environmental and biological factors, including food availability and temperature.

Conclusions

Studying the antioxidant defense in wild fish provides a basis of the oxidative state of the fresh product which is later subjected to post-mortem handling processes (storage, distribution, or heating during cooking) that promote additional oxidative damage. Results from this study suggest spatial–temporal variations of the oxidative stress indicators in muscle of totoaba that may be associated to a complex interaction between environmental (habitat type, potentially pollution) and biological (reproduction, nutrient availability) factors. Furthermore, these results contribute to explain the appeal of totoaba as a marketable meat and suggest totoaba may provide antioxidant nutrients to consumers.

Supplemental Information

Supplemental Information 1 Raw data

Click here for additional data file.

Authors acknowledge L Campos Dávila, NO Olguín Monroy, JJ Ramírez Rosas, L Rivera Rodríguez, F Valenzuela Quiñonez, O Rodríguez García, H Bervera León, M Román Rodríguez, M Vélez Alavez, J Isboset Saldaña, F Valverde, R Martínez Rincón and O Lugo Lugo for their assistance with sample collection and/or sample and data processing. Fishermen Federation from San Felipe and Golfo de Santa Clara provided assistance with field work.

Additional Information and Declarations

Competing Interests

Author Contributions

Animal Ethics

Field Study Permissions

Data Availability

The authors declare there are no competing interests.

Priscila Conde-Guerrero performed the experiments, analyzed the data, prepared figures and/or tables, authored or reviewed drafts of the paper, and approved the final draft.

Lia C Méndez-Rodríguez conceived and designed the experiments, analyzed the data, prepared figures and/or tables, authored or reviewed drafts of the paper, and approved the final draft.

Juan A. de Anda-Montañez conceived and designed the experiments, performed the experiments, analyzed the data, prepared figures and/or tables, authored or reviewed drafts of the paper, and approved the final draft.

Tania Zenteno-Savín conceived and designed the experiments, performed the experiments, analyzed the data, authored or reviewed drafts of the paper, and approved the final draft.

The following information was supplied relating to ethical approvals (i.e., approving body and any reference numbers):

Mexican government SGPA Dirección General de Vida Silvestre approved the study (SGPA/DGVS/02913/10, SGPA/DGVS/05508/11 and SGPA/DGVS/00039/13).

The following information was supplied relating to field study approvals (i.e., approving body and any reference numbers):

The Mexican government SGPA Dirección General de Vida Silvestre approved field collection permits SGPA/DGVS/02913/10, SGPA/DGVS/05508/11 and SGPA/DGVS/00039/13.

The following information was supplied regarding data availability:

Raw data are available as a Supplemental File.

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
