# Peer review of "Nutritional content of Totoaba macdonaldi (Gilbert, 1890), Antioxidants and lipid peroxidation in muscle"

_PeerJ, doi:10.7717/peerj.11129_

## Round 0.1 · original submission · Major Revisions

Reviewers have attended to the study, and found a lot of merit. However, some concerns have been raised, that need to be addressed.

Authors are encouraged to address all queries. Given that this is part 2 of a study, and part 1 appears to still be under review, it would be wise to consider this as part 1, since it deals with Antioxidants and lipid peroxidation in muscle. This is for authors to consider. It is also important that authors try to address this, and make this paper stand alone, and avoid reference to accompanying paper, as it appeared in this work.

At the beginning of materials and methods, it would also be appropriate to start it with subsection called 'overview of experimental study, comprise 3 sentences, providing a snapshot of the study, from 'collection of fish samples> Dissection & Freezing > Sample preparation for analytical studies > Laboratory analysis (x4) (Here, for each analysis, indicate the amount of samples allocated per analysis). Please, make sure that this is included in your revised manuscript.

Error, reference source not found, please kindly clarify.

Editor considers this work with great merit, and encourages authors to carefully and diligently revise it, with comments provided. Look forward to your revised manuscript. Thank you for your fine contribution to PeerJ.

Reviewer 1 ·

Basic reporting

Basic reporting is at the high level in the manuscript. Paper is well structured and quite well written, though some broken sentences do occur, and paper requires extensive proof reading for the minor typos and minor lexical improvement ( I marked some of the issues in the pdf attached).
I think that introduction should be restructured, to show the importance of studying oxidative stress and antioxidants, especially in application to totoaba. Now the introduction is quite disconnected from the research hypothesis. Also, since we are talking about totoaba here, I think it is important to mention connections between the totoaba fishing an expiration of the vaquita. Also please note that some references consistently failed to render properly (see annotated pdf).

Experimental design

I think design and statistical analysis are sound enough, but I would like to have more explanations regarding two following points:
1) why authors limited amount of the GLM's analysed is only 5 per enzyme? why not to run more models say with bootstrapping and then select the best ones based on AIC?
2) Please explain better how have you address unequal amount of the specimens in the different samples (i.e. from different ontogenetic stages or habitats. Also please check if all the raw data is attached.

Validity of the findings

I think findings of the paper are well supported by the analysis. I am especially excited about ecological and ecotoxicological implications of the paper, regarding the totoaba's oxidative stress reaction to the extraneous factors.

Additional comments

I would like to congratulate authors on a brilliant paper, dealing with such valuable and vulnerable species of the aquaculturaly important species. I think you done a great job showing ecotoxicological and environmental drivers behind the levels of the oxidative stress in totoaba. Please make larger emphasis on this part of your findings in the text (see pdf attached for the further comments).

Annotated reviews are not available for download in order to protect the identity of reviewers who chose to remain anonymous.

Reviewer 2 ·

Basic reporting

Basic reporting: The report was fairly written with good English not confusing to the reader. Introduction and background of the study were averagely well written and referenced to explain why and purpose of the research. Standard laboratory analytical methods were applied under materials and methods. Figures and Tables were fairly clearly displayed and presented, respectively. Raw data were also supplied.

Experimental design

Experimental design: The primary research was original and within the scope of the Journal. The major weakness of the study is experimental design. Further studies should be done with an acceptable design to discuss results better. This is also to assist in the repeatability of the research.

Validity of the findings

Validity of findings: Findings of the research were fairly valid based on the statistical analyses employed after data collection.
Conclusions require slight modifications depending on specific objectives of the study.

Additional comments

General comments: The authors analyzed and compared the antoxidant enzyme activities and lipid peroxidation levels in the muscle of totoaba, endemic species of fish found in Gulf California. Samples of the fish were collected from different locations and seasons along the route which the fish migrate to assess potential variation in nutritional content based on when or where the fish was caught. The paper is publishable but some issues as follows have to be addressed:
1. Title: Authors have to explain why title has reference and the way it was framed; why not as an instance: Effect of different locations and seasons on antioxidant enzyme activities and lipid peroxidation levels in the muscle of Totoaba Macdonaldi from Gulf California? Or Nutritional content of Totoaba Macdonaldi 2: Antioxidants and Lipid Peroxidation in the Muscle

2. Introduction: Objectives/aims written at the end of introduction requires modification. They could be listed as specific objectives (lines 76 to 79).

3. Materials and methods: Line 84: The statement with exclamation ‘Error-reference not found’ should be deleted or else authors would want readers to suggest references for them.

ii. What method of sampling was employed? Line 93-94 showed that gill net was used. Source of the net should be mentioned for the purpose of repeatability. Line 96-liquid nitrogen from where? Line 125: cited reference is better written once than repetition e.g. Perky et al. (Perky et al., 2000), Reference inside bracket should be enough
iii. Authors should explain why the study has no identifiable experimental design. The statistical analysis was quite complex!

4. Results and discussion: Authors tried to present and discussed the results and the findings fairly well. However, it would have been better with experimental design.

5. Conclusions: There should be slight modifications in the conclusion bearing in mind that conclusions from research studies are made considering the specific objectives and the various findings as discussed under results and discussion. Statement of lines 335-337 is a strong point under the conclusion.

6. References: Authors tried in the listing of the references for the report. However, three references were discovered not listed but cited inside text: Mala et al., 1997-line 42; Konietzko and Schwertz, 2016- line 50; Mexico’s….2001-line 46

Reviewer 3 ·

Basic reporting

No comment

Experimental design

no comment

Validity of the findings

no comment

Additional comments

General comments:
The results obtained are good and useful information and it is appropriate for publication.

Introduction: it is mentioned that there is a different paper regarding the nutritional aspect. So, i think this paper can be independent without reference to another unpublished paper. It is ok to state that there will be another paper regarding the nutrition value, but no refer to them since it's not published yet.
- line 51-52 “...however, information...is scarce” this sentence is not needed in this part, because the main study is in stress oxidative and enzymatic antioxidants.
- Line 56-57: “...thus, the aim of this study...of totoaba.” also the same. The aim of the study is to study the stress oxidative and antioxidants and it is already well mentioned in line 76.
- Line 58: content of trace element (accompanying paper). I think its not appropriate to cite unpublished paper, I suggest only focus on antioxidants and lipid peroxidation.

Materials and Methods:
- Sampling (line 82): change to “sample collection and preparation”
- Sample preparation: Join the sampling section. Is there any reference for this technique to prepare the muscle sample?
- Part of sampling explanation in statistical analysis (line 173-179) should be moved as first part of “sample collection and preparation”
- What about the replication of the analysis? Its not mentioned in the manuscript
- Wavelength of spectro. I think it would be better if the word “wavelength” is mentioned everytime the wavelength number is mentioned. For example: line 133 “... at 530 nm wavelength”.
- The 3 seasons sampling. It is mentioned that in summer, fish go deeper (line 68-69). Is this the main reason no sample was done in summer? If yes, I think it should be stated clearly in the sample collection section.
- Line 176: “Figure 1 accompanying paper” is not necessarily needed.

Results:
- Line 201, why aren't all genders found in all seasons? Are there any specific reasons?
- it is confusing and difficult to understand as a reader because they are mixed. it would be better if the results are separated and compared for each habitat in separate sections or subheadings:
Estuaries
Rocky area
Continental area
- line 222 and 226, k=9, if I understand it well, table 3 shows the k should be 10. please check and confirm

Discussion: well described

Conclusion: the conclusion is not in a good order. Conclusion should be made based on the results obtained.
- The first sentence can be a suggestion after conclusion is stated.
- Omega-6 HUFAs were not studied in this study, so its not necessary to refer this in conclusion
- Line 335 should be mentioned firstly.

Figure and table
- Table 2: It will be easier to read if the data is shown in mean with standard deviation. why the data are not shown in mean with the standard deviation? Is there any specific reason?
- Table 3: line 1 (Table 3) the tile should be "Table 3. Statistics of ..."

---

## Round 0.2 · Minor Revisions

Reviewers have attended to your revised manuscript. Acceptance was recommended by one, but not yet by another. Please, kindly attend to the concerns raised by one of the reviewers. Look forward to your feedback.
Thank you very much

Reviewer 2 ·

Basic reporting

No comments

Experimental design

The aspect of experimental design has been addressed.

Validity of the findings

The findings are valid.

Additional comments

No comments.

Reviewer 3 ·

Basic reporting

Clear and well-described

Experimental design

has been improved

Validity of the findings

well described

Additional comments

Thank you for the revision. Please find some minor comments. I hope it can improve the manuscript for better understanding for the reader.

Introduction:
Line 81: please add “which affect the quality of their muscle” AFTER “...biological factors throughout its lifespan.”
Line 85: “when and where” change to “season and location”

Materials and Methods:
Line 97: “Filed”, change to “field”
Line 132-190: please include the repetition in the analysis of parameters conducted, is it 2 replication or three replication.

Results, discussion and conclusion: well-described.

---

## Round 0.3 · accepted · Accept

The authors have sufficiently revised the manuscript, and it can now be accepted for publication. The Editor believes that authors have benefitted immensely from the peer-review process. Thank you for finding PeerJ as your journal of choice, and looking forward to your future works. Congratulations!